# Facts and Challenges in Immunotherapy for T-Cell Acute Lymphoblastic Leukemia

**DOI:** 10.3390/ijms21207685

**Published:** 2020-10-16

**Authors:** Fátima Bayón-Calderón, María L. Toribio, Sara González-García

**Affiliations:** Interactions with the Environment Program, Immune System Development and Function Unit, Centro de Biología Molecular Severo Ochoa, CSIC-UAM, 28049 Madrid, Spain; fbayon@cbm.csic.es

**Keywords:** T-cell acute lymphoblastic leukemia, immunotherapy, monoclonal antibodies, chimeric antigen receptor, relapse, leukemia-initiating cells

## Abstract

T-cell acute lymphoblastic leukemia (T-ALL), a T-cell malignant disease that mainly affects children, is still a medical challenge, especially for refractory patients for whom therapeutic options are scarce. Recent advances in immunotherapy for B-cell malignancies based on increasingly efficacious monoclonal antibodies (mAbs) and chimeric antigen receptors (CARs) have been encouraging for non-responding or relapsing patients suffering from other aggressive cancers like T-ALL. However, secondary life-threatening T-cell immunodeficiency due to shared expression of targeted antigens by healthy and malignant T cells is a main drawback of mAb—or CAR-based immunotherapies for T-ALL and other T-cell malignancies. This review provides a comprehensive update on the different immunotherapeutic strategies that are being currently applied to T-ALL. We highlight recent progress on the identification of new potential targets showing promising preclinical results and discuss current challenges and opportunities for developing novel safe and efficacious immunotherapies for T-ALL.

## 1. Introduction

T-cell acute lymphoblastic leukemia (T-ALL) is an aggressive hematological disorder that results from the progressive accumulation of genomic alterations in T-cell precursors developing in the thymus. T-ALL is characterized by the infiltration of bone marrow by immature T-cell lymphoblasts, while immature T-cell tumors characterized by a thymic mass and limited bone marrow infiltration are instead diagnosed as T-cell lymphoblastic lymphoma (T-LBL). T-ALL was described as an independent disease in the 1970s, after finding thymus-associated markers expressed on the surface of leukemic cells from pediatric patients [1]. Its incidence is higher in children than in adults (up to 25% and 15% of newly diagnosed ALL cases, respectively) [2], and it is twice as prevalent in males as in females [3]. Patients with T-ALL are frequently classified as high-risk due to the unfavorable features of the disease that include high leukocyte count, hematopoietic failure, and medullar and extramedullar infiltration with high probability of affectation of the central nervous system (CNS), which represents a frequent site of relapse [4]. In 1995, the European Group for Immunological Characterization of Leukemias (EGIL) established a classification of different clinically relevant T-ALL subtypes based on the expression of cell surface markers corresponding to sequential intrathymic T-cell developmental stages [5]: pro-T, pre-T, cortical and mature T-ALL (Figure 1). However, in recent years, improved genomic and transcriptomic techniques have provided new insights into the characterization of the prevalent genetic lesions involved in T-ALL pathogenesis [6,7,8], which has proved more valuable for risk-stratification of patients at the time of diagnosis [9,10,11]. Detection of translocations of enhancers or promoters of T-cell receptor (TCR) genes to other chromosomal regions helped the identification of the first T-ALL oncogenes, including the transcription factors *TAL1* [12], *LYL1* [13] and *TLX1/HOX11* [14]. Unique gene expression profiles were later found to be associated with leukemic arrest of thymocytes at different developmental stages [15], leading to the definition of new T-ALL subgroups characterized by the driver oncogenes or oncogene fusions (*TAL/LMO, TLX1, TLX3*, *HOXA*, and *MYB* genes), denoted as type A aberrations, present at diagnosis [16]. Other genetic alterations, denoted as type B, are recurrently detected in T-ALL patients and include point mutations, insertions and deletions (INDELs), and chromosomal gains or losses, which result in activation of the NOTCH1 T-cell fate specification pathway (*NOTCH1/FBXW7*) in more than 60% of T-ALL cases [17], activation of cytokine signaling pathways (*IL7R, JAK1/3, FLT3, CKIT, PI3K/AKT/PTEN, ABL1, N/KRAS*) and transcription factors (*RUNX1, ETV6, BCL11B, WT1, TCF7, LEF1, CTNNB1, GATA3, IKZF1*), inactivation of cell cycle inhibitors (*CDKN2A/B, CDKN1B, CDKN1C, CCND3, RB*), or deregulation of chromatin modifiers and remodeling factors (*PHF6*, *CTCF, KDM6A, SETD2, KMT2A/2D/2C, DNMT3A, IDH1/2*) (reviewed in [18,19,20,21,22]).

Nonetheless, molecular and genetic abnormalities were not considered for the characterization of T-ALL subtypes in the last revised edition of the World Health Organization (WHO) classification of ALLs in 2017 [23], and only early T-cell precursor lymphoblastic leukemia (ETP-ALL), which accounts for approximately 10% of pediatric and 40%–50% of adult T-ALL cases, was introduced as a new provisional entity [24]. ETP-ALL is characterized by the lack of T-lineage markers (preferentially CD1a, CD8 and weak CD5) but a phenotype that resembles that of early thymic progenitors (ETP), with expression of stem cell (CD34, CD117) and myeloid (CD13, CD33) lineage markers [25]. Genetically, ETP-ALL frequently associates with mutations in genes encoding epigenetic regulators (*IDH1, IDH2,* and *DNMT3A*), signaling factors (i.e., *NRAS* and *FLT3*), and transcription factors involved in hematopoietic and T-cell development (*RUNX1, GATA3* and *ETV6*), whereas both activating NOTCH1 mutations and *CDKN2A* deletions co-occurring with oncogenic NOTCH1 mutations are rarely observed [26,27]. ETP-ALL has been for years associated with poor prognosis [25,28,29], but application of early response-based intensification regimens in the last years has greatly improved the outcome of these patients [30,31].

In the 1960s, only 20% of T-ALL patients were cured, but nowadays, intensive chemotherapy as the standard front-line therapy for T-ALL has raised cure rates to above 85%. Current protocols for T-ALL patients include consecutive phases of induction, consolidation, delayed intensification, and maintenance, with drug combinations that commonly include doxorubicin or daunorubicin, dexamethasone or prednisone, vincristine, asparaginase, cyclophosphamide and cytarabine, together with methotrexate and intrathecal chemotherapy as prophylaxis for CNS infiltration [32,33,34]. In a retrospective study, the Children’s Oncology Group (COG) reported that 5 yr overall survival (OS) for patients younger than 20 years who enrolled in their ALL clinical trials increased from 70.7% in 1990–1994 to 81.6% in 2000–2005 [35]. Similar 5 yr disease-free survival (DFS) and OS (83.8% and 89.5%, respectively) were obtained for all children and young adults (1 to 31 yr) enrolled in the AALL043 methotrexate early-intensification study by the same group from January 2007 to July 2014 [36]. However, adult T-ALL still presents a dismal outcome, with significantly lower survival rates than pediatric T-ALL. Although 90%–95% of adult patients achieved complete remission (CR) in different trials [37,38,39], OS after 3 and 5 years was only 65% and 48% respectively, with percentages decreasing with age to only 27% 5 yr OS for patients aged over 50 years. Relapse occurred in 30%–40% of adult T-ALL patients within the 7–24 months after remission and less than 10% of the relapsing patients survived [36,37]. Minimal residual disease (MRD) at the end of the induction phase is the key prognostic factor of relapse. MRD assessment in childhood T-ALL, either by real time quantitative polymerase chain reaction (PCR) detection of TCR gene rearrangements or by flow cytometry immunophenotyping of leukemic cells, has established MRD ≥ 10^−3^ as the most important predictive factor of relapse [40,41]. In adult T-ALL, MRD level ≥ 10^−4^ is associated with higher incidence of relapse and reduced OS, and has been a criteria used to classify high-risk patients [42,43].

The therapeutic available options for patients experiencing relapse or for those who are refractory to standard chemotherapeutic regimes are very scarce, and since the approval of nelarabine by the US Food and Drug Administration (FDA) in 2005 [44], no new agents have been specifically developed for T-ALL. This is certainly not the case for relapsed and/or refractory (r/r) B-cell acute lymphoblastic leukemia (B-ALL) patients, whose life expectancy has increased considerably in the last years after the introduction of anti-CD22 antibodies, bi-specific T-cell engagers (BITEs) and, lately, chimeric antigen receptors (CARs). Although nelarabine, a cytotoxic DNA damaging agent, has improved the survival of T-ALL relapsing patients [45,46,47], its dose-limiting toxicity [48,49,50], together with the absence of alternatives, underscore the need for new targeted therapies. However, the shared expression of surface markers between normal and leukemic T cells has limited the development of new targeted immunotherapies against T-cell malignancies and particularly, against T-ALL. This is due to the induction of secondary T-cell immunodeficiency is associated with therapy, which may result in the appearance of opportunistic infections and/or the reactivation of latent infections leading to life-threatening situations. Consequently, main challenges of future T-ALL treatments rely on (1) the identification of unique markers of T-ALL blasts, especially of those expressed on leukemia-initiating cells (LICs), which are the drivers of relapse [51], and (2) the elucidation of therapies aimed at killing leukemic but not healthy T cells, in order to avoid immunodeficiency.

In this review, we will discuss recent immunotherapy strategies based on monoclonal antibodies (mAbs) and CARs that are currently being tested in clinical trials for T-ALL (Table 1 and Figure 2), including molecular mechanisms, preclinical studies and expected clinical applications. We will also focus on novel molecules (Figure 2) emerging as promising T-ALL therapeutic targets in preclinical mouse models and discuss how their use is anticipated for clinical immunotherapy.

## 2. Immunotherapy of T-ALL with Monoclonal Antibodies

### 2.1. Mechanisms Underlying Monoclonal Antibody Therapy

Different strategies of the immune system directed to combat pathogens have been the basis for the development of cancer immunotherapy. Regarding mAbs, their effectiveness as immunotherapeutic agents is based on the ability to recognize antigens expressed on the cell surface, and the capacity to either block the interaction with ligands or inhibit receptor clustering and activation, thereby leading to apoptosis of target cells. In addition, mAbs activate effector mechanisms that lead to the elimination of antigen-expressing cells [53]. These mechanisms include antibody-dependent cellular cytotoxicity (ADCC) and complement-dependent cytotoxicity (CDC). ADCC implies the binding of mAbs to cell surface antigens and the recognition of the mAb Fc portions by specific receptors [54], expressed mainly on macrophages and natural killer (NK) cells [55]. This recognition stimulates the cytotoxic activity of the immune cell, leading to the release of granzyme and perforin that eventually induce the killing of target cells [56]. In CDC, the complement cascade is activated through different pathways after the recognition of cell-surface bound mAbs, inducing the formation of the membrane attack complex (MAC) and leading to cell lysis [57]. mAb immunotherapy targeting unique surface antigens expressed by tumor cells uses mechanisms like ADCC and CDC to induce tumor cell death (specific examples are described in the following sections).

An alternative mAb-based immunotherapy relies on antibody-drug conjugates (ADCs), consisting of mAbs conjugated to cytotoxic agents that are internalized after antigen recognition, releasing the coupled toxin into the intracellular space and inducing apoptosis on target cells. Initial ADCs based on mAbs developed in mouse showed low target specificity and produced high immunogenicity when applied to humans. Later, development of antibody humanization techniques and conjugation to more potent drugs contributed to the approval in 2000 by the FDA of the first ADC (Gemtuzumab ozogamicin), raised against the CD33 molecule for acute myeloid leukemia (AML) treatment. By 2020, more than 80 different antibodies have been approved for their use in different pathologies [58,59]. In this section, we will focus on the different antibodies that are currently under investigation in clinical trials for T-ALL (Table 1), discussing the main challenges they have still to face for their introduction as first-line therapies.

### 2.2. CD38

CD38 is a type II transmembrane glycoprotein [60] that acts as a catalytically active transporter [61], whose function and localization on lipid rafts depend on tetramerization [62]. It serves both as cyclase and as hydrolase in the synthesis and degradation, respectively, of cyclic ADP-ribose (cADPR) [63]. cADPR acts then as a second messenger both in Ca^2+^ mobilization from the endoplasmic reticulum [64], and in Ca^2+^ entry from the extracellular compartment [65]. CD38 is expressed at early stages of B- and T-cell development but is lost in mature naïve lymphocytes [66,67], being re-expressed after T-cell activation [68,69], and again lost in the T-cell memory compartment [70]. In T cells, CD38 signaling is associated with TCR functionality, triggering the activation of intracellular molecules including Zap70, phospholipase C gamma (PLCγ), Raf1, Erk and Akt/PKB [71,72,73]. Activation of CD38 through recognition of its ligand, CD31, has been shown to induce death in developing T cells and was suggested to contribute to thymocyte selection in the thymus [74]. CD38 also participates in adhesion events during the migration of lymphocytes through the endothelial cell wall [75], and was shown to regulate cytotoxic T-cell activity in vitro [76].

The wide expression of CD38 in hematological malignancies, including multiple myeloma (MM), chronic lymphocytic leukemia (CLL), ALL, AML, and NK-cell leukemias (reviewed in [77]), has made this antigen an attractive molecule for immunotherapeutic strategies. Anti-CD38 antibodies were proved effective in preclinical mouse models targeting CLL [78] and MM [79,80] in preclinical mouse models. Daratumumab is a human immunoglobulin G1 kappa (IgG1K) that binds with high affinity to a unique epitope of human CD38, which induces cell killing mediated by CDC, ADCC and antibody-dependent cellular phagocytosis (ADCP) [79]. Different studies using immunodeficient mice have demonstrated its activity even in the absence of a competent immune system, suggesting an underlying mechanism of action still to be understood. Either as monotherapy [81] or in combination with other agents [82], Daratumumab showed great efficacy in patients with MM, and was approved in 2015 by the FDA for the treatment of refractory MM. Other anti-CD38 mAbs, Isatuximab (approved in 2020 by the FDA for MM treatment) and MOR03087/MOR202, are currently being tested in different trials for other pathologies. The safety and promising efficacy of anti-CD38 mAb-based treatments has prompted the design and application of CAR T-cell therapies targeting CD38, either as a single antigen or in combination with other antigens (Multi-CAR), which are still in progress for r/r AML (NCT03222674, NCT03473457, NCT04351022), MM (NCT03464916, NCT03767751, NCT03778346, NCT03473496) and B-ALL (NCT03754764, NCT03125577, NCT04430530) patients.

Robust expression of CD38 in different T-ALL subtypes has recently been reported [83], with no drastic expression changes associated with chemotherapy. This suggests that CD38 targeting could be a valid approach also for T-ALL patients who relapse or do not respond to conventional therapies, and proof of concept was, in fact, obtained by different studies in animal models. CD38 upregulation can be induced in CD38low adult T-cell leukemia (ATL) cells by treatment with all-trans retinoic acid (ATRA), a clinically accepted compound used to treat patients with acute promyelocytic leukemia (APL), thus exposing leukemic cells to recognition by CD38 CAR T cells [84]. Interestingly, encouraging results have recently been communicated about the use of Daratumumab for T-ALL in preclinical studies [85]. CD38 was detected both in ETP-ALL and in non-ETP-ALL patient leukemic cells. Using T-ALL patient-derived xenograft (PDX) models, it was shown that almost all PDXs (14 out of 15) responded to Daratumumab, with remarkable positive outcomes for ETP-ALLs. By mimicking T-ALL induction phase-like chemotherapy in PDX preclinical models, Vogiatzi et al. demonstrated the effectiveness of Daratumumab in eliminating MRD, either alone or in combination with chemotherapeutic compounds [86]. Moreover, Daratumumab administration to a 19-yr old patient with refractory ETP-ALL was well tolerated and produced a temporary reduction to less than 1% of T-cell lymphoblasts in the bone marrow [87]. Daratumumab also eliminated MRD in three T-ALL patients suffering relapse after allogeneic stem cell transplantation [88].

Given the advanced status of anti-CD38 antibodies in the clinics, two different clinical trials are now testing the efficacy of Daratumumab (NCT03384654) and Isatuximab (NCT03860844) for r/r B- or T-ALL (Table 1). In case positive results are obtained from these ongoing trials, CD38 CAR T-cell therapy could be also envisioned for T-ALL patients in the near future.

### 2.3. CD52

CD52 is a small membrane-bound glycoprotein of 12 amino acids that anchors to glycosylphosphatidylinositol (GPI) [89]. CD52 is widely expressed in the hematopoietic compartment, including mature lymphocytes, eosinophils, dendritic cells, monocytes/macrophages, and neutrophils, but is absent in erythrocytes, platelets, hematopoietic progenitors, and in immature T cells (reviewed in [90]). CD52 function is relatively unknown, but current evidences indicate that it can act as a stimulatory molecule for T-cell activation and generation of regulatory T cells (Treg) from CD4^+^ T cells [91,92]. Recently, human and mouse CD52^hi^ CD4^+^ Treg cell populations have been identified, in which PLC releases a soluble isoform of CD52 that binds to Siglec-10 on activated T cells, inducing a Toll-like receptor (TLR)-mediated immunosuppressive mechanism [93,94].

CD52 antigen has extensively been used as a target of several immunotherapeutic approaches after the development of CAMPATH-1 [95], a rat mAb that was used in vitro to deplete mature lymphocytes from allogeneic human bone marrow grafts as a strategy to avoid graft-versus-host disease (GvHD) in bone marrow transplants. The humanized mAb (CAMPATH-1H) is the basis of Alemtuzumab, which was approved by the FDA in 2001 for the treatment of r/r CLL [96]. Upon binding to CD52, Alemtuzumab induces cell killing by CDC, ADCC and apoptosis of target cells [89,97,98]. Beyond CLL, CD52 expression has been described in AML, different subtypes of mature T/NK neoplasms [99], B-ALL, and some cases of T-ALL [100,101]. However, both untreated pre-B and pre-T leukemic blasts seem to express lower levels of CD52 than their healthy mature counterparts at the time of diagnosis [102], suggesting that CD52 expression is not a common feature of ALL and might be restricted to more mature subtypes.

Different clinical trials have tested the efficacy of Alemtuzumab for T-cell malignancies such as cutaneous T-cell lymphoma (CTCL), T-cell prolymphocytic leukemia (T-PLL), T-cell large granular lymphocyte leukemia (T-LGL), ATL and T-ALL [97,103,104,105]. Complete and partial responses (CR and PR) were observed in different patients, particularly those suffering from T-PLL, with manageable toxicity that was attributed mainly to viral infections following lymphopenia. However, none of the T-ALL patients achieved a response [105]. In a later trial, adult B- and T-ALL patients that presented more than 10% of CD52^+^ lymphoblasts were subjected to Alemtuzumab in combination with chemotherapeutic drugs in an effort to eradicate MRD [106]. Although MRD was reduced in several patients, myelosuppression, lymphopenia and toxicity associated with viral reactivation of cytomegalovirus-, Herpes simplex- or Herpes zoster-latent infections were communicated, which prompted the termination of these trials. Only one study has reported the use of CAMPATH-1H as single-therapy in r/r pediatric B- and T-ALL patients [107]. One patient (1/13) with standard-risk B-precursor ALL (BCP-ALL) showed CR, resulting in an objective response rate of 8% (95% confidence interval 0.2%–36%). Although patients tolerated CAMPATH-1H relatively well, with only two patients referring grade IV pain and grade III allergy/hypersensitivity, trial was terminated before stage 2 due to poor accrual. Even though these antecedents pointed to a relative effectiveness of anti-CD52 mAb therapy in eliminating MRD in T-ALL, no clear benefits have been observed compared to other available therapies, and no new trials targeting CD52 have been initiated for r/r T-ALL recently. However, evidences from different studies highlighted the effectiveness of Alemtuzumab in eliminating lymphocytes from peripheral blood and bone marrow. Accordingly, CD52 immunotherapy has been widely applied to allogeneic stem cell grafts prior transplantation to prevent GvHD [108]. The characteristics of CD52 function have also been exploited for the development of CAR T cells in which the CD52 gene is knocked out by TALEN technology, rendering them insensitive to anti-CD52 therapy [109]. This approach allows for depletion of autologous patient’s T cells using Alemtuzumab prior CAR T-cell infusion, therefore enhancing CAR T-cell engraftment and maintenance [110,111]. Preclinical assays also suggest that CAR T cells can be efficiently eliminated after tumor eradication in xenograft models of AML using Alemtuzumab as a bridge to allow hematopoietic stem and progenitor cell (HSPC) transplantation [112].

## 3. Immunotherapy of T-ALL with CAR T Cells

Over the past few years, immunotherapy based on adoptive transfer of T cells engineered to express CARs against tumor antigens has emerged as a powerful strategy in the treatment of refractory hematopoietic malignancies such as B-ALL [113]. Considering the similarities between B- and T-lymphoid malignancies, broadening CAR T-cell therapy to the latter could seem straightforward. However, CAR T-cell therapy against T-cell malignancies, although feasible [114,115], still presents some limitations, which have delayed its clinical application. Firstly, the shared expression of target antigens between CAR T cells and malignant T cells can lead to CAR T-cell fratricide by self-recognition, resulting in impaired antitumor efficacy. Additionally, as pointed out above, malignant and normal T cells share a similar profile of surface protein expression, thus, the on-target/off-tumor cytotoxicity of CAR T-cell therapy can cause severe T-cell aplasia, leading to a life-threatening immunodeficiency.

Briefly, CAR T-cell immunotherapy consists of the genetic modification of the patient’s T cells in order to recognize and kill cancer cells. T cells are extracted from the patient’s peripheral blood and modified ex vivo to express a receptor that binds to a certain surface molecule present in cancer cells. In this way, CAR T cells are able to target and eliminate tumor cells in an antigen-specific and major histocompatibility complex (MHC)-independent manner. The genetically engineered T cells are then expanded in vitro and infused back into the patient, where, upon recognition of the specific tumor antigen, they will become activated and will direct their cytotoxic activity against tumor cells.

CARs are synthetic hybrid receptors that comprise an extracellular target-binding moiety, such as a single chain variable fragment (scFv), and a transmembrane domain (TM) and a hinge domain, which anchor the receptor to the cell membrane and protrude the scFv into the extracellular space, respectively (Figure 3). In addition, CARs contain intracellular signaling modules that activate T-cell effector functions following antigen engagement. The sequences of the hinge domain are generally derived from segments of IgG subclasses (such as IgG1 and IgG4), IgD, CD8α, CD28, CD4 or CD3 molecules [116], while the most commonly used TM domains derive from CD8α, CD28 or CD4 [117]. The intracellular region is composed of signaling domains from the TCR-CD3 complex as well as from costimulatory molecules like CD28, 4-1BB, OX40 or ICOS. The different components that conform the intracellular region define the different generations of CARs. First-generation CARs contain a single signaling domain corresponding to the CD3ζ (CD247) molecule, which is part of the TCR-CD3 complex and regulates the intracellular signaling downstream of TCR [118]. This first generation of CARs, although able to activate and induce cytotoxicity against target cells, could not sustain long-term proliferation and persistence of CAR T cells [119]. Thus, second- and third-generation CARs were designed, which incorporated one or more intracellular domains from T-cell costimulatory molecules, providing necessary signals for eliciting T-cell activation and for achieving robust CAR T-cell expansion and persistent antitumor activity [120]. Although the clinical application of CARs has mostly focused on second- and third-generation CAR structures, fourth- and fifth-generation CARs are also being tested in preclinical models. Fourth-generation CAR T cells, also known as TRUCKs (T cells Redirected for Universal Cytokine Killing), are T cells engineered to co-express a CAR together with an antitumor cytokine intended to recruit endogenous immune cells that mediate tumor eradication. The expression of this cytokine might be either constitutive or induced by T-cell activation upon antigen engagement. Fifth-generation CAR T cells refer to universal CAR T cells (uCAR-T), which are genetically engineered to no longer express endogenous TCR and/or MHC molecules in order to avoid GvHD or graft rejection, respectively. This strategy would benefit those patients who are not suitable for autologous CAR T-cell therapy, particularly those suffering from T-cell malignancies [121].

Given the success of CAR T-cell therapies for B-cell malignancies (reviewed in [122,123]) and the promising preclinical results described when targeting T-cell antigens with mAbs, several trials have been initiated using CAR T cells against different types of T-cell leukemias/lymphomas, including T-ALL (Table 1). Below, we summarize the findings that led to the development of these CAR T cells and how their clinical application is envisioned.

### 3.1. CD5

CD5 is one of the most common surface markers of malignant T cells, expressed in around 80% of T-ALLs and T-cell lymphomas [124,125]. In normal cells, its expression is restricted to thymocytes, peripheral T cells, and to a minor subpopulation of B cells (B-1 cells) [126,127]. This transmembrane receptor regulates T-cell development and function, acting as a negative regulator of TCR-CD3 signaling [128,129]. The association of CD5 with the TCR raises the threshold required to activate a T cell, rendering it tolerant to its cognate antigens [130]. Thus, CD5 prevents T cells from uncontrolled self-reactivity, playing a fundamental role in protecting against autoimmunity. CD5 has also been described as an activation marker of T cells [131,132]. It is upregulated at crucial check-points of thymocyte development following pre-TCR and TCR activation, its expression being directly related to the strength of the signal transmitted by these two receptors [132]. CD5 was validated as a tumor target antigen in clinical trials using toxin-conjugated CD5 mAbs. These clinical trials showed depletion of cancer T cells in T-ALL and cutaneous T-cell lymphoma patients without severe adverse effects related to on-target/off-tumor activity [133,134,135], thereby suggesting that the development of CAR T-cell therapy targeting this widely expressed T-cell marker could be safely used in the treatment of T-cell malignancies. In this context, Mamonkin et al. [115] reported that T cells transduced to express a second-generation CD5 CAR using the CD28 costimulatory domain showed CD5 downregulation, resulting in limited and transient fratricide and allowing its expansion ex vivo. Moreover, these CD5 CAR T cells were able to effectively eliminate T-ALL and T-cell lymphoma lines in vitro and to control disease progression in xenograft mouse models of T-ALL. An ongoing clinical trial (NCT03081910) for patients with r/r T-ALL and T-cell lymphoma is evaluating the safety and efficacy of this approach.

To overcome the fratricide observed among CAR T cells due to inherent CD5 expression, some groups have suggested the use of CD5-negative cells, such as NK cells [136] and CD5-CRISPR-Cas9-edited T cells [137], as CAR-modified effector cells for the targeting of T-cell malignancies. Wada et al. [138] analyzed T cells transduced with lentivirus containing third-generation CD5 CARs that incorporated CD28 and 4-1BB as costimulatory domains. These CD5 CAR T cells potently lysed CD5^+^ malignant T-cell lines and primary tumors in vitro at higher or comparable rates than previous reports, while significantly controlling tumor expansion and improving survival in T-ALL xenograft models. However, this construct also showed a high degree of cytotoxicity against normal T cells. Subsequently, an inducible safety switch based on the human CD52 gene was incorporated into the CD5 CAR construct. Thereby, by administration of Alemtuzumab, which specifically binds to CD52 and induces cell death via CDC and ADCC as described above, authors observed a specific depletion of CD5 CAR T cells in mouse peripheral blood and spleen. Similar safety switch strategies based on the co-expression of surface molecules such as CD20 or truncated human epidermal growth factor receptor (tEGFR) have been proposed. Administration of specific anti-CD20 or anti-tEGFR mAbs, respectively, eliminates CAR-T cells once tumors are eradicated and avoids severe adverse effects due to long-term persistence of CAR-expressing cells. Other strategies consist on the co-expression of the CAR together with suicide switches based either on metabolic-mediated mechanisms such as Herpes simplex virus thymidine kinase (HSV-tk) or dimerization-inducing chimeric proteins like inducible caspase 9 (iCasp9) [139,140]. Further preclinical studies are necessary in order to evaluate the use of inducible safety switching as a mechanism for modulating CAR T-cell activity and expansion.

### 3.2. CD7

CD7 is a TM glycoprotein highly expressed in lymphoblastic T-cell leukemias and lymphomas (>95%) and in a subset of peripheral T-cell lymphomas (PTCL) [125,141]. It is also expressed by the majority of peripheral T cells and NK cells and their precursors, playing an essential role in T-cell activation and interactions with other immune cell subsets [142]. However, it does not appear to play a pivotal role in T-cell development or function, since genomic disruption of CD7 in murine T-cell progenitors allows for unperturbed development and homeostasis with minor alterations in T-cell effector function [143,144].

CD7 was evaluated as a target for immunotoxin-loaded mAbs in patients with T-cell malignancies [145]. In that study, severe CD7-related toxicities were not observed, but tumor responses were relatively modest. Regarding CD7 CAR T-cell therapy, different groups reported that, unlike CD5, the downregulation of CD7 on transduced CAR T cells is incomplete, which leads to fulminant fratricide and precludes ex vivo expansion [146,147,148]. Therefore, surface expression of CD7 must be disrupted in CAR T cells in order to eliminate persistent self-antigen targeting in CAR T cells. Abrogating CD7 expression either by gene editing [146,147] or by blocking CD7 protein trafficking to the cell surface [148], minimized fratricide and allowed expansion of CD7 CAR T cells without affecting proliferation or short-term effector function. Consequently, a robust antitumor activity in preclinical studies against primary CD7^+^ T-ALL and lymphoma has been observed in preclinical studies. A Phase I clinical trial (NCT03690011) is under development for the evaluation of CD7-CRISPR-Cas9-edited CD7 CAR T cells treatment of patients with CD7^+^ T-cell malignancies, including T-ALL.

One step further was taken by Cooper et al. [147], who described a strategy for the treatment of r/r T-ALL and non-Hodgkin’s T-cell lymphoma based on a third-generation (CD28 and 4-1BB costimulatory domains) CD7-specific uCAR. CRISPR/Cas9 genetically edited human T cells lacking both CD7 and TRAC (T-cell receptor alpha chain) were used as the source for generating of CAR T cells (CD7 uCAR-T). Since deletion of TRAC results in impaired TCR-mediated signaling, this approach would not only prevent fratricide, but would also allow for the use of allogeneic CAR T cells without inducing life-threatening GvHD. CD7 uCAR T cells demonstrated efficacy against human T-ALL cell lines and patient-derived primary T-ALL both in vitro and in vivo, without fratricide or T-cell-mediated xenogenic GvHD. Although these findings must be deeply evaluated before translating into the clinic, they hold promise for the development of a feasible adoptive T-cell therapy for T-cell malignancies without a requirement for autologous T cells.

### 3.3. CD3

CD3 is a multimeric (CD3ε, CD3γ, CD3δ, CD3ζ) protein complex widely expressed at variable intensity levels on the cell surface of mature T-cell lymphomas and mature T-ALLs [149], whereas cytoplasmic CD3 expression is a mandatory diagnostic marker for immature T-ALL subtypes. In normal cells, CD3 expression is limited to the hematopoietic system and occurs in association with the TCR or the pre-TCR, specifically on the surface of T cells and thymocytes. TCR-associated CD3 molecules have been, for years, the focus of different therapeutic approaches for tolerance induction in autoimmune diseases, and for preventing organ rejection. Indeed, OKT3, a mouse anti-CD3ε mAb (Muromonab), was the first antibody to become available in humans to prevent allograft rejection after transplantation of solid organs. Although therapy with anti-CD3 antibodies has been mainly directed to induction of Treg cells [150], some reports have pointed to the therapeutic application of these mAbs in T-ALL. Anti-CD3ε mAb treatment of mice transplanted with either murine or human T-ALL cells was shown to cause massive leukemic blasts death, irrespective of the underlying genetic alterations, and to trigger a molecular process that resembles negative selection normally occurring during T-cell development in the thymus [151].

Similarly to CD5 and CD7 molecules, CD3 was firstly evaluated as a therapeutic target in patients with T-cell lymphoma in studies using immunotoxin-loaded anti-CD3ε mAbs [152]. This treatment, although short-lived, was well tolerated, causing partial remissions in some patients and thus leading to the development of anti-CD3 CAR-engineered cells. First efforts in this regard focused on using NK cells as CAR-expressing cells, since they lack surface CD3 [153]. Expression of a third-generation CD3 CAR, containing CD28 and 4-1BB costimulatory domains, in the NK-92 cell line was shown to specifically eliminate CD3^+^ lymphoma cell lines and primary T cells in vitro, as well as CD3^+^ Jurkat T cells in xenograft murine models [153]. Additionally, Rasaiyaah et al. [154] reported a strategy for manufacturing CD3 CAR T cells harboring TALEN-mediated disruption of endogenous TCRαβ/CD3 complex. This approach, based on a second-generation anti-CD3ε CAR that incorporates a 4-1BB costimulatory domain, showed specific cytotoxicity against CD3^+^ primary T cells and childhood T-ALL samples in vitro. These CD3 CAR T cells also mediated potent antileukemic effects in a human/murine chimeric model. Notably, a second-generation (4-1BB costimulatory domain) CD3 CAR has recently been used to develop a novel strategy aimed at generating allogeneic T cells lacking TCRαβ for CAR T-cell therapies [155]. The early and transient expression of a second-generation (4-1BB costimulatory domain) anti-CD3 CAR in donor T cells, shortly after TCRα/CD3 disruption via TALEN, programs CAR T cells to self-eliminate the remaining TCRαβ positive cells, obtaining an ultrapure TCRαβ negative cell population at the end of the CAR T-cell manufacturing process. The production of these universal CAR T cells did not alter proliferation, differentiation or exhaustion properties of T cells, neither affected the cell killing capacity of the allogeneic engineered uCAR-T cells. Therefore, although the potential of this technology should be deeply examined in extensive in vivo studies, it seems an attractive alternative to autologous T-cell transfer.

### 3.4. CD4

CD4 was one of the first molecules studied as a target for CAR T-cell immunotherapy, since it is highly expressed in a significant proportion of mature T-cell lymphomas and in a subset of T-ALLs [156]. In normal cells, CD4 is expressed not only in a large proportion of thymocytes (80%–90%) and over 50% of peripheral blood T cells (helper, memory and regulatory subsets) [157,158], but also in several non-T cell hematopoietic populations such as monocytes, macrophages, granulocytes and Langerhans cells [159,160,161]. CD4 acts as a co-receptor for the TCR to initiate or augment the early phase of T-cell activation that follows recognition of antigens displayed by antigen-presenting cells (APCs) in the context of MHC-II molecules (reviewed in [162]).

Clinical trials using anti-CD4 mAbs showed that depletion of CD4^+^ cells is reversible and well tolerated without evidence of immunosuppression in many cases [163,164,165,166]. These results encouraged Pinz et al. to develop a third-generation CD4 CAR (containing 4-1BB and CD28 costimulatory domains), which showed preclinical efficacy in vitro and in a xenograft mouse model of PTCL, when both T cells [156] and NK-92 cells [167] were used as CAR carriers. Currently, a phase I clinical trial (NCT04162340) has been initiated to evaluate the safety and antitumor efficacy of CD4 CAR T cells against CD4^+^ T-cell hematological malignancies, including T-ALL. Ma et al. [168] extended these studies using the same CD4 CAR structure but incorporating a natural safety switch based on CAMPATH (Alemtuzumab) to treat CD4^+^ T-ALL. Treatment with Alemtuzumab, which recognizes CD52 on CAR T cells, would eliminate infused CD4 CAR T cells after tumor depletion, thus preventing toxicities associated with CD4^+^ cell aplasia caused by long-term persistence of CAR T cells. In this assay, CD4 CAR T cells were able to efficiently and potently target malignant CD4^+^ T-ALL cells in murine models, showing robust antitumor effects in vivo. The equivalent to a low human dose of CAMPATH quickly and efficiently depleted the infused CD4 CAR T cells, which may be crucial in the clinics for patient safety. Further studies should be conducted to compare the susceptibility to different dosages of Alemtuzumab and the appropriate timing of treatment leading to elimination of the vast majority of CAR T cells without compromising their antitumor efficacy.

### 3.5. CD1a

CD1a is a member of the CD1 family of transmembrane glycoproteins structurally related to the MHC proteins, which mediates presentation of self- or microbial-derived lipid and glycolipid antigens to specialized T cells [169,170]. CD1a is found on around 40% of T-ALL cases, defining the cortical T-ALL subtype (coT-ALL) [171,172]. CD1a expression on normal tissues is restricted to developing cortical thymocytes, skin Langerhans cells and a subset of circulating myeloid dendritic cells, being absent on mature T cells [173,174,175,176]. Addressing the in vivo role of CD1a has long remained a difficult task due to the fact that, while there are four isoforms of CD1 in humans (CD1a, CD1b, CD1c and CD1d), only the CD1d isoform is expressed in the mouse [169]. Nevertheless, a recent study using CD1a transgenic mice reported that the expression of this molecule is responsible for the pathogenesis of poison-induced dermatitis and psoriasis [177]. This work also showed that treatment with CD1a-blocking antibodies mitigated skin inflammation without comorbidity or side effects, supporting CD1a as a safe target for coT-ALL treatment. In this regard, Sánchez-Martínez and colleagues [173] recently focused their efforts on the development of second-generation (4-1BB costimulatory domain) CD1a CAR T cells that were resistant to self-antigen-driven fratricide without the need of knocking out the target gene on effector cells, as CD1a expression is absent on peripheral blood T cells. In this study, CD1a CAR T cells showed robust and specific cytotoxicity against CD1a^+^ coT-ALL cell lines and primary coT-ALL cells, both in vitro and in murine xenograft models. A phase I clinical trial is intended to confirm the efficacy and safety of a CD1a-directed CAR T-cell therapy for r/r coT-ALL patients.

### 3.6. TCRβ Constant Region

The TCRαβ is a pan-T cell marker responsible for T-cell activation after recognition of antigen-derived peptides bound to MHC molecules expressed on APCs. It is a heterodimer composed of two variable protein chains (TCRα and TCRβ) associated with invariant CD3 molecules. TCRβ chains contain variable (V), diversity (D), joining (J), and constant (C) regions, and TCRα only V, D and C regions. TCR diversity is based on somatic recombination of individual V, D, and J segments, which leads to T-cell clonality as a result of expression of the same unique rearranged TCR in a particular T-cell population. The TCRαβ is present in >95% of cases of PTCL, almost all angioimmunoblastic T-cell lymphomas (AITL) and around 30% of T-ALLs [178,179]. Taking advantage of the mutually exclusive expression of TCRβ-chain constant domains 1 and 2 (TRBC1 and TRBC2) on mature T cells, and of the fact that about half of TCR^+^ T-cell lymphomas solely express TRBC1, Maciocia et al. [180] recently developed an original approach for targeting mature T-cell cancers. They engineered third-generation (CD28 and OX40 costimulatory domains) anti-TRBC1 CAR T cells that were proved to recognized and killed normal and malignant TRBC1^+^ but not TRBC2^+^ T cells in vitro, and in a disseminated mouse model of leukemia. This strategy could thus eradicate mature T-cell malignancies while preserving approximately two-thirds of the normal T-cell population in patients. A phase I clinical study is now evaluating the safety of this approach against TRBC1^+^ T-cell lymphomas (NCT03590574), and it could be extended in the near future to T-ALL patients.

## 4. Potential New Targets for T-ALL Immunotherapy

The identification of antigens with an expression restricted to malignant T cells has been the unsuccessful goal pursued by numerous investigations whose final aim is the development of suitable immunotherapy strategies that preserve the pool of healthy T cells. Development of powerful bioinformatics techniques in the last years has allowed the identification of few antigens that seem to be specific of leukemic T cells, not expressed by normal mature T cells nor by HSPCs [181,182]. Yet, the effectiveness of these molecules as immunotherapeutic targets remains to be investigated. Meanwhile, several surface proteins, despite being shared between leukemic T-cell, some normal T-cell subpopulations, and other non-T cell lineages, have emerged as critical players in T-ALL pathogenesis. Moreover, some of them, including CXCR4, CD44, and interleukin-7 receptor (IL-7R), have recently been validated as promising therapeutic targets in T-ALL preclinical animal models (Figure 2).

### 4.1. CXCR4

CXCR4 (C-X-C chemokine receptor type 4, CD184) is a seven-transmembrane domain G-protein-coupled receptor of the chemokine receptor family that interacts with CXCL12 (SDF-1). CXCR4 is expressed in several cell types including HSPCs, where it plays an important role in the maintenance of stem cell interactions with components of the bone marrow [183,184]. CXCR4 signaling is also essential for B lymphopoiesis [185,186] and for T-cell migration to different organs, including CNS [187,188,189,190,191].

The involvement of the CXCR4/CXCL12 signaling axis in the pathogenesis of several types of cancers was expected considering its implication in both cell migration and regulation of interactions with components of the tumor microenvironment, a function that may ultimately regulate metastasis [192,193]. The essential role of CXCR4 in T-ALL pathogenesis was recently demonstrated in a NOTCH1-induced T-ALL mouse model [194], where inhibition of endogenous CXCR4 expression by short hairpin RNA (shRNA) impaired CXCL12-induced migratory properties of T-ALL cells, and also induced cell death and altered cell cycle progression in vitro. When transplanted into immunodeficient mice, both murine and human CXCR4-silenced T-ALL samples showed delayed leukemia onset accompanied by increased host survival, as compared to control shRNA-transduced cells. This effect was mediated by reduced LIC frequencies and a deficient bone marrow homing of CXCR4-silenced T-ALLs. In vivo, T-ALL cells are found in close contact with bone marrow CXCL12-producing cells and this interaction is crucial for disease progression. In fact, CXCR4 targeting leads to remission of established murine T-ALLs and human PDXs [195]. In a different study, pharmacological CXCR4 antagonists inhibited bone marrow colonization by T-ALLs and reduced the neuropathologies associated with CNS infiltration [196], suggesting that CXCR4 may be a potential therapeutic target to eradicate CNS affectation in T-ALL patients. Several reports have suggested a relationship between Notch signaling and CXCR4 in T-ALL, although no direct regulation of CXCR4 expression by NOTCH has been documented. In a Notch1-induced T-ALL mouse model, bone marrow T-ALL cell colonization reduced the wild type HSPC pool, suppressed B-cell development, and caused thrombocytopenia, thus altering normal hematopoiesis [197]. These effects were mediated by a competition of T-ALL cells for perivascular regions and by the activation of Notch signaling in stromal cells, which led to a reduction in osteoblastic cells and a negative regulation of CXCL12 transcription. Therefore, T-ALL-mediated disruption of bone marrow niches promotes malignant progression at the expense of normal hematopoiesis through CXCR4/CXCL12 deregulation. Deregulated Notch3 activation also results in CXCR4 expression in preleukemic CD4^+^CD8^+^ intrathymic T cells that migrate to and colonize the bone marrow in a CXCR4-dependent manner [198].

While no differences in *CXCR4* mRNA expression were detected between leukemic and normal T cells in microarray analysis [199], elevated CXCR4 surface expression was reported in T-ALL leukemic blasts compared to normal peripheral T cells, both in human and mouse [195]. This would be an advantage for implementation of CXCR4-targeted therapies. However, double-positive (DP) thymocytes express high levels of CXCR4, which raises important concerns about potential induction of thymic T-cell aplasia by CXCR4 inhibitors. Up to date, several small-molecule CXCR4 antagonists have been developed (reviewed in [200]), which mainly act by inhibiting CXCL12 binding and have been used for HSPCs mobilization from bone marrow [201,202]. Among them, Plerixafor (AMD3100) was approved by the FDA in 2008, in combination with G-CSF, for mobilization of HSPCs to peripheral blood prior to autologous stem cell transplantation. It has also been used for r/r AML [203,204] and CLL treatment showing promising results [205]. In a different trial, children with r/r AML, ALL or myelodysplastic syndrome (MDS) were treated with Plerixafor in combination with cytarabine and etoposide [206], and although AML patients achieved complete remission, no clinical response was observed in ALL and MDS patients. Another CXCR4 antagonist, POL5551, was shown to increase cytarabine sensitivity in a xenograft model of a high-risk pediatric ALL subtype, infant MLL-rearranged (MLL-R) ALL [207], indicating that disruption of the CXCR4/CXCL12 interaction may benefit children with high-risk ALL.

Based on the promising results obtained with pharmacological CXCR4 antagonists, several anti-CXCR4 antibodies have been produced and tested in preclinical models against different types of leukemia [208,209,210], and some of them have entered into clinical trials. Ulocuplumab (BMS-936564/MDX1338) was shown to block CXCL12 binding to CXCR4-expressing cells, inhibiting CXCL12-induced migration and inducing apoptosis in different hematological malignancies [211]. In a recent phase Ib/II study in adult patients with r/r MM, Ulocuplumab in combination with lenalidomide and dexamethasone resulted in a high response rate, although neutropenia and thrombocytopenia were observed in some patients [212]. Additionally, LY2624587 antibody was capable of inhibiting tumor growth in a CCRF-CEM T-ALL xenograft model by preventing CXCR4 signaling [213]. Therefore, given the potential antileukemic effects of CXCR4 inhibition in preclinical T-ALL assays, and the acceptable tolerance to different chemical inhibitors and antibodies already tested in patients, it may be worth expanding these studies on CXCR4/CXCL12-axis targeting to r/r T-ALL patients. Currently, there is an open phase IIa trial to assay the CXCR4 antagonist BL-8040 (NCT02763384) in combination with nelarabine in adult patients with r/r T-ALL/T-LBL.

### 4.2. IL-7R

IL-7R is a type I heterodimeric transmembrane receptor for IL-7, consisting of an α chain (IL-7Rα) and a common gamma chain (γc) that is shared between several cytokine receptors (IL-2, IL-4, IL-9, IL-15 and IL-21). IL-7Rα can also heterodimerize with the CRLF2 (cytokine receptor-like factor 2) subunit to generate a high affinity receptor for thymic stromal lymphopoietin (TSLPR) [214]. IL-7 was firstly described as a growth factor for developing B cells [215], but its role in cell survival and proliferation was quickly proved essential also for T cells [216,217]. IL-7-induced IL-7R signaling is indispensable to accomplish the full intrathymic maturation program of both gamma-delta (γδ) and alpha-beta (αβ) T cells [218,219,220,221]. It regulates the first expansion wave that bone marrow-derived progenitors seeding the thymus must undergo in order to generate the immature T-cell pool, which will afterward rearrange *TCR* genes and develop into either TCRγδ or TCRαβ mature T cells. IL-7 also plays a relevant role in the maintenance and survival of naïve [222,223] and memory peripheral T cells [224,225,226,227,228], as well as in innate lymphoid cell development [229,230]. Given that T-ALL arises from the malignant transformation of immature T-cell progenitors, it was not unforeseen that a high proportion (more than 70%) of T-ALL cases express functional IL-7Rs that mediate proliferative and prosurvival signals in response to IL-7 [231,232,233,234]. More recently, a possible role of IL-7R in T-ALL molecular pathogenesis started to become apparent after discovering that IL-7R is a direct target of NOTCH1 and is involved in Notch-mediated T-ALL cell maintenance [235]. Since the report in 2004 that *NOTCH1* is a commonly mutated gene in T-ALL [17], deciphering the molecular mechanisms behind NOTCH1 deregulation leading to normal T-cell transformation, and elucidation of NOTCH1 target genes involved in leukemia cell growth, have been the keystones for developing novel targeted therapies for T-ALL patients harboring NOTCH1 mutations.

NOTCH1 signaling induces IL-7R expression by transcriptional upregulation of the *IL7R* gene (encoding IL-7R) through binding of the CSL/MAML-1 transcriptional complex to the *IL7R* promoter [235], a mechanism conserved in the mouse [234]. In addition, genome-wide mapping of NOTCH1 DNA-binding sites has highlighted a major role of super-enhancers in the dynamic regulation of NOTCH1 target genes including *IL7R* [236]. These findings indicate that the cell growth-promoting effects of the NOTCH1 oncogenic program are enhanced by means of *IL7R* gene induction and activation of IL-7R signaling, pointing to IL-7R downstream effectors as promising molecular targets for therapeutic intervention in T-ALL. As in normal T cells, IL-7R signaling in T-ALL cells is triggered by binding of IL-7 to, and heterodimerization of, IL-7Rα and γc chains, which places the intracellular domains of the receptor and the attached JAK1 and JAK3 kinases into close contact. This initiates serial phosphorylation events that activate STAT proteins (STAT1, STAT3 and STAT5) and PI3K/Akt/mTOR and MEK/Erk signaling pathways, regulating the expression of downstream effectors such as SOCS, Cyclin D2, BCL-2 and p27kip1 (reviewed in [237]). In T-ALL, PI3K activation is mandatory for IL-7-mediated p27kip1 down-regulation, Rb hyperphosphorylation, and BCL-2 upregulation, and is also required for cell size increase, expression of the glucose transporter Glut1, glucose uptake, and maintenance of mitochondrial integrity [238]. These findings indicate that PI3K is a major effector of IL-7–induced viability, metabolic activation, growth, and proliferation of T-ALL cells, highlighting the importance of developing targeting interventions directed to the IL-7R pathway. Because some B-ALL cases express functional IL-7Rs and respond to IL-7 as well [231,239], therapeutic targeting of IL-7R could be extended to B-cell leukemias. Several drugs capable of targeting IL-7R-activated routes are beginning to be explored clinically, with special emphasis on PI3K/Akt/mTOR, JAK/STAT5, and BCL-2 inhibition (reviewed in [32,240]).

Besides the accepted role of IL-7R in the maintenance and progression of T-ALL, several studies have pointed to a direct contribution of this signaling axis in the pathogenesis of T-ALL. Lymphoid malignancies develop spontaneously in both IL-7-transgenic and AKR/J mice that naturally overexpress IL-7Rα [241,242], and increased membrane IL-7R density leading to enhanced IL-7 signaling has been shown to cooperate with T-cell oncogenes in mouse T-ALL [243]. Conversely, progression of T-ALL patient-derived xenografts is hampered in IL-7-deficient mice [244]. More importantly, up to 10% of T-ALL cases harbor activating mutations in genes encoding different molecular effectors downstream of IL-7R signaling, including JAK1, JAK3 and STAT5B (reviewed in [199]), reinforcing the role of IL-7R in leukemia cell proliferation. Recently, direct proof has been provided that IL-7R signaling does, in fact, contribute to T-ALL pathogenesis [234]. Using loss-of-function approaches, it was formally established that IL-7R signaling is essential for Notch1 oncogenicity and T-cell leukemogenesis. The work also demonstrated that IL-7R expression contributes to human T-ALL LIC function, and revealed that IL-7R is a functional biomarker of T-ALL cells with LIC potential, a finding that was extended to B-ALLs expressing IL-7R [234]. The realization that IL-7R/IL-7 signaling directly contributes to T-ALL LIC activity has important clinical implications, as new therapeutic developments against leukemia mostly rely on molecular targets of LIC cells, which are finally responsible of disease relapse [51,245]. Therefore, targeting of normal IL-7R signaling represents a promising therapeutic approach for preventing T-ALL and, likely, B-ALL relapses, the major hurdle of ALL [246].

Overall, accumulated findings have placed IL-7R into focus as an oncogenic receptor for T-ALLs. This possibility was later confirmed by seminal studies showing that *IL7R* gain-of-function somatic mutations occur in around 10% of T-ALL and 1% of BCP-ALL patients at diagnosis [247,248]. These mutations are located in exon 5, and mostly in exon 6 encoding the juxtamembrane-transmembrane region of IL-7Rα. Exon 6 mutations consist of short insertions/deletions that introduce an unpaired cysteine residue capable of disulphide bond formation, allowing for pairing of two IL-7Rα chains and generation of an α-α homodimer, which triggers the activation of downstream IL-7R signaling cascades independently of IL-7. Decoding the signaling pathways selectively activated by oncogenic IL-7R, as compared to those induced by wild type IL-7R, will provide strong rationale for tailoring T-ALL (and B-ALL) therapies based on targetable IL-7R-related molecules that specifically eliminate leukemic cells, while preserving normal T-cell development and homeostasis. Yet, identification of such exclusive oncogenic molecules/pathways remains challenging. Thus, targeting of IL-7Rα itself has emerged as a promising alternative strategy for a great majority of T-ALL patients and some B-ALL patients, even for those harboring *IL7R* activating mutations.

Recent advances in clinical immunotherapy have now reached the IL-7R pathway. Several anti-IL-7Rα blocking mAbs are currently in Phase I or Phase IIa trials in patients with autoimmune diseases where IL-7R seems to play a critical role (reviewed in [249]), such as type 1 diabetes and multiple sclerosis, or Sjogren’s syndrome and inflammatory bowel disease, respectively (reviewed in [250]). In T-ALL, PDX models based on administration of anti-IL-7Rα mAbs and ADCs have provided promising results for therapeutic targeting of T-ALLs expressing either wild type or mutant IL-7Rs [251,252], and preclinical studies have also proved that mAb-mediated IL-7R targeting impairs B-ALL progression and metastasis [253]. Recently, Barata and co-workers [251] have generated a fully human mAb that recognizes human IL-7Rα and impairs IL-7/IL-7R-mediated signaling. The antibody sensitized T-ALL cells to treatment with dexamethasone, induced leukemia cell death in vitro through NK cell-mediated ADCC, and reduced leukemic burden in vivo. Conjugation with the antineoplastic toxin monomethyl auristatin E (MMAE) significantly increased the leukemia killing capacity of the mAb, validating the ADC approach. Additional work by Durum and coworkers showed that anti-IL-7Rα mAb-mediated ADCC had therapeutic efficacy against both relapsing and established disease in T-ALL preclinical models [252], leaving the door open for clinical trials evaluating the efficacy of this therapeutic strategy in unresponsive and relapsed patients.

An alternative, yet unavailable, strategy of targeting refractory T-ALLs expressing either wild type or mutant IL-7R may rely on the administration of anti-IL-7Rα CAR T cells. This approach is particularly challenging, considering the IL-7R-dependency of the lymphoid compartment and the finding that IL-7R is expressed in non-hematopoietic tissues as well [254,255]. This represents an important drawback, as potential therapeutic benefits may be counteracted by associated off-target toxicities and severe combined immunodeficiency [256]. Nevertheless, recent studies suggest that immunosuppression induced by treatment of mouse autoimmune arthritis with an anti-IL-7Rα ADC could be well tolerated clinically for a period of time [257], and very promising results have been reported from anti-IL-7R mAb clinical trials for autoimmune diseases [258]. Overall, current information on the major contribution of normal IL-7R signaling to the onset, maintenance, and progression of acute leukemias opens a new and major area of research to develop and validate novel therapies focusing on the IL-7R pathway.

### 4.3. CD44

CD44 is a type I single-span transmembrane glycoprotein that functions as a receptor for the glycosaminoglycan hyaluronan (hyaluronic acid, HA) [259,260], a major component of the extracellular matrix (ECM). The *CD44* gene contains 19 exons in humans, and generates a variety of tissue-specific isoforms different in size through alternative splicing and *N—*and *O-*glycosylation [261,262,263]. In the hematopoietic system, the CD44 standard isoform (CD44s), containing 10 exons, is the most abundant, whereas variant isoforms (CD44v), which contain different exons assembled in the variable domain at the juxtamembrane region of the receptor, present a more restricted expression [264]. CD44v isoforms are expressed in discrete populations of epithelial cells and in some hematopoietic cell subsets, particularly during development; in several types of carcinoma [265], and after lymphocyte activation [266]. Many studies have focused on the CD44v6 isoform because it is frequently upregulated in cancer cells of epithelial origin, where it plays a role in migration, metastasis and chemoresistance (reviewed in [267]) and has been associated with poor prognosis [268,269].

The first described function for CD44 was as a molecule implicated in lymphocyte homing into lymph nodes [263]. Afterward, CD44 has been shown to play a prominent role in the anchoring of both hematopoietic precursors [270,271] and leukemic cells [272] within the bone marrow niche. Notably, CD44 is also expressed during intrathymic T-cell development, both in human and mouse, where it defines different progenitor stages with singular intrinsic linage potentialities. CD44 is expressed at high levels in the earliest human CD34^+^ T-cell precursors seeding the thymus, and its expression is downregulated upon T-cell commitment and during T-cell development, while it is maintained in myeloid-primed intrathymic progenitors with dendritic cell potential [273,274,275].

Importantly, CD44 is one of the most studied markers of cancer-initiating cell (CIC), with a key function in the maintenance of CIC-associated properties in several solid tumors (reviewed in [267]). However, its contribution to LIC activity in hematologic cancers, although suggested [272], has not been formally proven until very recently. By using a novel mouse model of human T-ALL pathogenesis [276], this work showed that CD44 is a direct transcriptional target of NOTCH1, which is upregulated in NOTCH1-induced preleukemic blasts, facilitating their engraftment into the bone marrow niche, and supporting the LIC potential of T-ALL [276]. These results indicate that CD44 upregulation is a key event of T-cell leukemogenesis, a finding that concurs with the fact that T-ALL patients commonly display an aberrant expression of CD44 in leukemic blasts [277]. Although no link of CD44 expression with prognosis and overall survival of T-ALL patients has been established [277], CD44 contributes to T-ALL maintenance and progression, likely by regulating LIC potential [278], as proved by anti-CD44 mAb administration in a PDX T-ALL model [276]. CD44 expression is also upregulated in Notch1-induced T-ALL leukemic cells treated with chemotherapeutic drugs, such as doxorubicin and dexamethasone, and contributes to T-ALL chemoresistance by modulating intracellular drug efflux [279]. Consequently, CD44 may represent a valuable target for new therapies against relapsed or chemoresistant T-ALLs.

Promising results have been obtained in r/r AML patients treated with an anti-CD44 blocking antibody [280]. In PDX preclinical settings, grafting of AML patient cells into mouse bone marrow can be blocked by administration of an activating anti-CD44 mAb [281], which targets AML leukemic stem cells by inducing their differentiation [282] or by increasing apoptosis [283,284], leading to AML eradication. An ADC consisting of an anti-CD44 antibody (Bivatuzumab) bound to mertansine was tested in phase I clinical trials for head and neck and esophagus squamous cell carcinomas [285,286] and for breast cancer [287], but the trials had to be terminated before progression to phase II due to severe skin toxicities. Recent trials have also tested a commercialized anti-CD44 mAb developed by Roche (RO5429083) for metastatic and/or advanced solid tumors and AML but, although completed, the trial results have not been published yet. Besides mAbs, recent development of anti-CD44 CARs has opened new opportunities for treatment of CD44-expressing cancers. In particular, anti-CD44v6 CAR T-cell immunotherapy trials for multiple cancers (NCT04427449), r/r AML, MM (NCT04097301) and breast cancer (NCT02046928) are ongoing, representing promising strategies worth to be extended to T-ALL and other CD44^+^ T-cell malignancies, once safety concerns are solved and efficacy proved.

### 4.4. Other Potential Targets: CD30 and CD99

CD30 belongs to the tumor necrosis factor receptor (TNFR) superfamily [288] and, although initially described as an antigen expressed on Hodgkin´s lymphoma (HL) cells, it was later found to be expressed in other hematological malignancies such as anaplastic large cell lymphomas (ALCL), CTCL, PTCL, adult T-cell leukemia/lymphoma, and diffuse large B cell lymphomas (DLBCL) [289,290]. In normal cells, CD30 is expressed by T helper (Th) cells [291] (preferentially Th2-type cytokine secreting cells [292]), by a subset of CD8^+^ peripheral blood lymphocytes [293], and by the B-1 subset of B lymphocytes [294]. Upregulation of CD30 expression was documented after stimulation or viral infections [i.e., Epstein-Barr virus (EBV), human T-cell leukemia virus (HTLV) and human immunodeficiency virus (HIV)] in lymphoid cells. Although CD30 induction was described after activation of both HSPCs [295] and T cells [296], the expression levels observed were lower than those displayed by tumor cells. Consequently, anti-CD30 treatment has proven to be effective in eliminating tumor cells without affecting normal lymphopoiesis in preclinical assays, which pointed to CD30 as a very attractive candidate for immunotherapeutic targeting. Anti-CD30 immunotherapy has been successfully applied as ADC (brentuximab vedotin) to r/r adult T-cell leukemia/lymphoma patients [297], and r/r CD30^+^ lymphoma patients [298], showing an especially good response in Hodgkin lymphoma and systemic ALCL cases. In addition, anti-CD30 directed CAR T cells have been tested in clinical trials for r/r HL or ALCL, reporting acceptable tolerability but modest effectiveness [299,300]. To date, several trials are open to assess safety and efficacy of CD30 CAR T cells in different hematological malignancies, although none of them include T-ALL patients. However, CD30 expression (>20% positive cells as assessed by flow cytometry) has been detected in 13/34 cases of T-ALL and in 6/44 cases of B-ALL [301]. Furthermore, CD30 expression is increased in bone marrow samples of T-ALL patients treated with high-dose chemotherapy, indicating that application of anti-CD30 immunotherapy should be considered when designing new trials for CD30^+^ r/r T-ALL patients.

CD99 (MIC2) is an *O*-glycosylated transmembrane protein expressed on leukocytes and activated endothelium that mediates cell adhesion [302], T-cell activation [303], and transendothelial migration of different cell types, including leukocytes [304]. The *CD99* gene encodes two different isoforms with differential expression patterns and functions. Anti-CD99 monoclonal antibodies were able to induce caspase-independent cell death of AML cell lines and primary blasts [305], T-cell lines [306] and BCP-ALL cells bearing the TEL/AML1 fusion [307]. T-ALL cells have been shown to express higher levels of CD99 than hematopoietic stem cells and normal T cells [308,309], and detection of CD99 expression by flow cytometry was demonstrated useful for revealing MRD in T-ALL patients [308,310], pointing to CD99 as a promising target for immunotherapy. However, LIC frequency assays indicated that CD99^+^ T-ALL fraction was not enriched in cells with LIC potential [309], thus precluding the suitability of CD99 as therapeutic target for relapsed T-ALL. A potential new therapeutic option based on CD99 targeting has emerged from in vitro studies showing that treatment of T- and B-cell lines with anti-CD99 antibodies leads to upregulation of heat shock protein 70 (HSP70) (a prerequisite for NK-dependent lytic activity), making leukemic cells susceptible to NK-cell cytotoxicity [311]. Thus, reassessment of CD99 as a candidate for alternative therapeutic strategies against T-ALL seems realistic.

## 5. Conclusions and Remarks

Recent advances in targeted immunotherapies for B-cell malignancies have engendered unprecedented expectations for the successful treatment of T-ALL patients, with the challenge still pending on establishing protocols for clinical management of associated side effects. Nevertheless, current progress, particularly on 1) the application of universal off-the-shelf CAR T cells that prevent fratricide; 2) the incorporation of suicide genes allowing for CAR T-cell elimination after tumor eradication and T-cell immunodeficiency reversion; and 3) the discovery of increasingly specific molecular targets proved critical for disease progression in preclinical models, has tilted the balance between risks and benefits towards the use of immunotherapy for r/r T-ALL. Still, avoidance of associated adverse effects demands further efforts for the identification of new unique T-ALL antigens absent on normal T cells that guarantee the safe and effective application of these strategies.

## Figures and Tables

**Figure 1 ijms-21-07685-f001:**
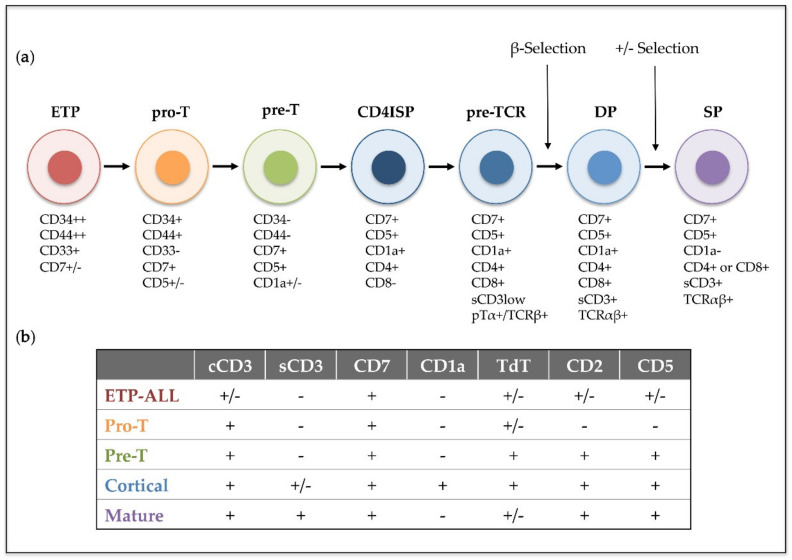
Human intrathymic T-cell developmental stages and corresponding T-ALL phenotypic subtypes. (**a**) Early thymic precursors (ETP) that have entered the human thymus differentiate through sequential developmental stages: pro-T, pre-T, immature CD4 single positive (CD4ISP), pre-T-cell receptor (pre-TCR), double positive CD4^+^ CD8^+^ (DP) and single posive (SP), which can be identified by the expression of different cell surface markers and the sequential acquisition of the pre-TCR and the TCRαβ receptors. (**b**) T-ALL subtypes (ETP-ALL, Pro-T, Pre-T, Cortical and Mature) are based on the 1995 EGIL classification of acute leukemias and the 2017 WHO classification of Tumours of Haematopoietic and Lymphoid Tissues, which included ETP-ALL as a new provisional entity. This classification was established by immunophenotyping based on surface molecules expressed at sequential intrathymic stages of human T-cell development. cCD3 (cytoplasmic CD3); sCD3 (surface CD3); TdT (terminal deoxynucleotidyl transferase). Presence or absence of the indicated molecule is represented by (+) and (−), respectively.

**Figure 2 ijms-21-07685-f002:**
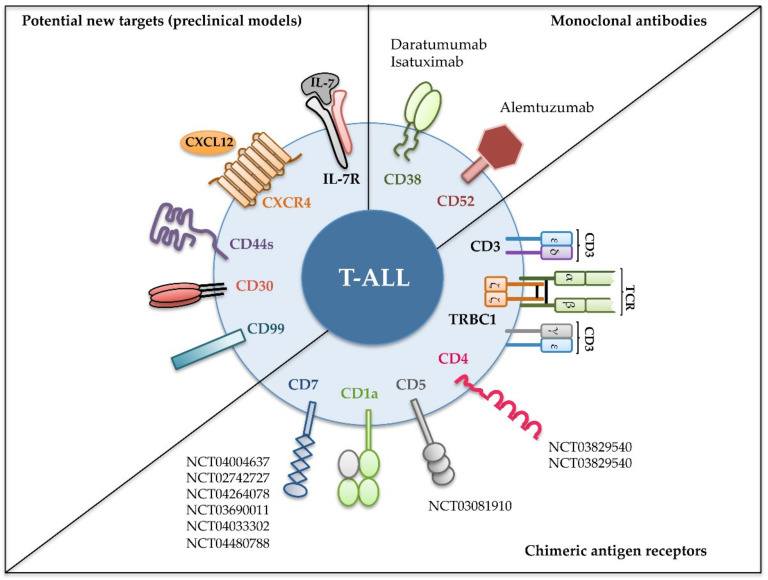
Recent immunotherapeutic interventions for T-ALL. Different strategies targeting the indicated cell surface molecules expressed on leukemic cells have been designed for T-ALL treatment. Monoclonal antibodies have been tested in the clinic against relapsed and/or refractory T-ALL (upper right); chimeric antigen receptors were assayed either in preclinical or in clinical settings (lower right); and new molecules have revealed their potential as promising targets in preclinical models (left). Identifiers [The National Clinical Trial (NCT) number] of clinical trials in progress including T-ALL patients are shown [52].

**Figure 3 ijms-21-07685-f003:**
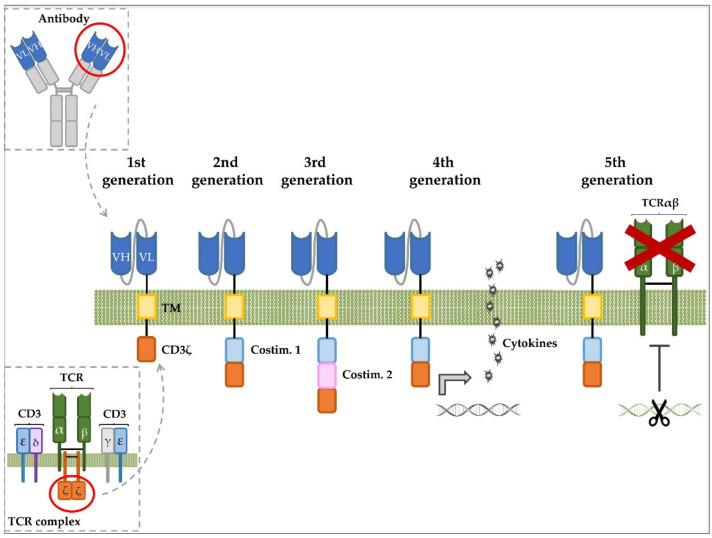
Schematic representation of the different CAR generations. CARs are conformed by an extracellular antigen recognition domain derived from the VH and VL antibody regions, a transmembrane domain (TM) and an intracellular signaling domain. First-generation CARs contain a single intracellular signaling domain derived from the CD3ζ chain of the TCR complex (CD3ζ), whereas second- and third-generation CARs incorporate additional costimulatory molecules (Costim.1 or Costim.2), derived from signaling molecules such as CD28 or 4-1BB. Fourth-generation CAR T cells co-express the CAR together with an immune system stimulatory cytokine, which is expressed either constitutively or after activation of a downstream transcription factor following antigen engagement. Fifth-generation CAR T cells express a second-generation CAR, and are genetically engineered to no longer express endogenous TCR and/or MHC molecules.

**Table 1 ijms-21-07685-t001:** Immunotherapy clinical trials [52] based on monoclonal antibodies and chimeric antigen receptors for T-cell malignancies, including T-ALL patients. * as by 28 August 2020.

**Monoclonal Antibodies**					
**Identifier**	**Disease**	**Intervention**	**Age of Patients**	**Phase**	**Status ***
NCT02999633	T-ALL/T-LBL	Isatuximab (anti-CD38)	16 yr and older	Phase II	Terminated
NCT00199030	T-ALL/T-LBL	Alemtuzumab (anti-CD52)	18 yr and older	Phase II	Completed
NCT00061048	Adult T-cell leukemia	Alemtuzumab (anti-CD52)	18 yr and older	Phase II	Completed
NCT03384654	T-ALL/B-ALL	Daratumumab (anti-CD38)	11 to 30 yr	Phase II	Recruiting
NCT00061945	B-/T-precursor lymphoblastic leukemia	Alemtuzumab (anti-CD52)	15 yr and older	Phase I/II	Completed
NCT03860844	T-ALL/B-ALL/AML	Isatuximab (anti-CD38)	Up to 17 yr	Phase II	Recruiting
**Chimeric Antigen Receptors**					
**Identifier**	**Disease**	**Intervention**	**Age of patients**	**Phase**	**Status ***
NCT03829540	CD4 + T-cell leukemia	CD4 CAR T cells	18 yr and older	Phase I	Recruiting
NCT04162340	CD4 + T-cell malignancies	CD4 CAR T cells	18 yr and older	Phase I	Recruiting
NCT03081910	CD5 + T-cell malignancies	CD5 CAR T cells	Up to 75 yr	Phase I	Recruiting
NCT04004637	CD7 + NK/T lymphoma; T-LBL; T-ALL	CD7 CAR T cells	7 to 70 yr	Phase I	Recruiting
NCT02742727	CD7 + r/r leukemia or lymphoma	CD7 NK cells	18 yr and older	Phase I/II	Unknown
NCT04264078	CD7 + NK or T-cell malignancies	Universal CD7 CAR T cells	2 to 70 yr	Early Phase I	Not yet recruiting
NCT03690011	CD7 + T-cell leukemia/lymphoma	CD7 KO-CD7 CAR T cells	Up to 75 yr	Phase I	Not yet recruiting
NCT04033302	CD7 + hematological malignancies	CD7 CAR T cells	6 mo to 75 yr	Phase I/II	Recruiting
NCT04480788	CD7 + r/r hematological malignancies	CD7 CAR T cells	7 to 70 yr	Phase I	Not yet recruiting

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
