# Peer review of "Facts and Challenges in Immunotherapy for T-Cell Acute Lymphoblastic Leukemia"

_ijms, 2020, doi:10.3390/ijms21207685_

Round 1
Reviewer 1 Report
In this manuscript, Bayon-Calderon et al review the current knowledge of immunotherapy-based treatments for T cell Acute Lymphoblastic Leukemia. The authors reflect on major challenges in the field, i.e. shared antigen expression between normal cells and cancer cells, the risks associated with interventions targeting such antigens, and the need to identify unique tumor antigens. Life-threatening reactivity of modalities that target such shared antigens has been a learning lesson for the immunotherapy field and has guided thorough examinations of candidate molecules to ensure safety, in addition to efficacy.
The review is well written, and I only have minor edits with regard to certain formatting and typing errors in the manuscript:
- “identified” (not indentified) in Line 62
- “established” (not stablished) in Line 67
- Line 129: Figure 2 missing (partially visible on page 5)
- Line 149-150: Transition from description of mechanisms in previous line, to conclude “Therefore, abundant and unique surface antigens… immunotherapy.” seems a bit abrupt. A suggestion for changing this is, “mAb immunotherapy targeting unique surface antigens expressed by tumor cells use mechanisms like ADCC and CDC to induce tumor cell death (Specific examples described in following sections)”.
- Line 392: “TCR activation, its expression being directly related” (instead of “being its expression directly related”)
- Line 809: “NCT” (not NTC)
Author Response
Point-by-point reply to Reviewer’s comments: (highlighted in red)
REVIEWER 1
In this manuscript, Bayon-Calderon et al review the current knowledge of immunotherapy-based treatments for T cell Acute Lymphoblastic Leukemia. The authors reflect on major challenges in the field, i.e. shared antigen expression between normal cells and cancer cells, the risks associated with interventions targeting such antigens, and the need to identify unique tumor antigens. Life-threatening reactivity of modalities that target such shared antigens has been a learning lesson for the immunotherapy field and has guided thorough examinations of candidate molecules to ensure safety, in addition to efficacy.
The review is well written, and I only have minor edits with regard to certain formatting and typing errors in the manuscript:
1. “identified” (not indentified) in Line 62
“indentified” has been corrected to “identified” in Line 62.
2. “established” (not stablished) in Line 67
“stablished” has been corrected to “established” in Line 67.
3. Line 129: Figure 2 missing (partially visible on page 5)
Figure 2 is now shown in page 5. It seems that Table 1 was split during the editing process. A request has been made to the editor for Table 1 to appear in its entirety on a single page (new page 4).
4. Line 149-150: Transition from description of mechanisms in previous line, to conclude “Therefore, abundant and unique surface antigens… immunotherapy.” seems a bit abrupt. A suggestion for changing this is, “mAb immunotherapy targeting unique surface antigens expressed by tumor cells use mechanisms like ADCC and CDC to induce tumor cell death (Specific examples described in following sections)”.
As suggested by the Reviewer, the sentence “Therefore, abundant and unique surface antigens expressed by tumor cells represent optimal targets for mAb immunotherapy” in old Lines 149-150, has been replaced by “mAb immunotherapy targeting unique surface antigens expressed by tumor cells use mechanisms like ADCC and CDC to induce tumor cell death (Specific examples described in following sections)” (new Lines 149-151).
5. Line 392: “TCR activation, its expression being directly related” (instead of “being its expression directly related”)
Following Reviewer’s suggestion, the sentence “TCR activation, being its expression directly related to…” in old Line 392 was changed to “TCR activation, its expression being directly related to...” (new Line 347).
6. Line 809: “NCT” (not NTC)
“NTC” has been changed to “NCT” in new Line 752 (old Line 809).
Reviewer 2 Report
The authors in the manuscript entitled “Facts and Challenges in Immunotherapy for T-cell Acute Lymphoblastic Leukemia” reviewed different therapeutic strategies currently pursued for T-ALL.
The authors described the progress of mAbs and CAT T cell-based therapies for T-ALL in the clinic, highlighting the challenges and opportunities associated with the development of safe and efficacious treatment options for T-ALL.
The authors discussed both the validated and the new targets in T-ALL and the mechanisms by which mAbs and CAR Ts are being exploited to combat T-ALL.
The manuscript is well written, well organized and different aspects were described appropriately.
Author Response
We thank the reviewer for their comments. No further changes have been included.